# Rethinking Developmental Curricula for Contrastive Visual Learning

## Abstract

While large machine learning models have achieved remarkable results, they still fall short of the efficiency and adaptability characteristic of human perception. Motivated by prior work that draws inspiration from infant visual development, we examine whether commonly used developmental proxies confer measurable benefits under controlled experimental conditions. Within a virtual environment, we systematically modulated four dynamic factors, namely image blur, lighting complexity, avatar movement speed, and scene complexity, as developmentally inspired components of a learning curriculum. However, none of these factor-wise curricula improved downstream classification performance compared with a stable training corpus. We then replicated the experiments on the real-world SAYCam dataset, varying movement speed and scene complexity independently, and observed consistent results. These findings suggest that, under the present training regime and evaluation suite, these particular factor-wise curricula do not inherently confer learning advantages. More broadly, the results contextualize claims that developmental-like progression inherently benefits learning and highlight the need for more principled curriculum design mechanisms. Our results offer a new perspective on curriculum design for self-supervised learning.

## 1 Introduction

Although large machine learning models have shown impressive performance and are increasingly integrated into daily life (Brown et al., 2020; Kirillov et al., 2023; Rombach et al., 2022), they continue to fall short of matching the efficiency and adaptability of human perception. Most contemporary models rely on vast datasets and substantial computational resources to achieve high performance; yet, as dataset sizes increase, their marginal performance gains diminish (Shetty & Siddiqa, 2019; Kaplan et al., 2020). Moreover, such models often exhibit limited generalization and robustness to environmental variations, domains in which human cognition naturally excels. Infants, for instance, learn to recognize and encode complex stimuli through a remarkably efficient and adaptive process that requires neither extensive data nor heavy computation (Saffran et al., 1996; Kellman & Garrigan, 2009). This gap indicates fundamental limitations in current machine learning methodologies and has motivated growing interest in exploring whether principles derived from human development can inform machine learning.

Curriculum learning has been proposed as a mechanism to improve training efficiency by structuring examples in a meaningful progression from simple to more complex (Bengio et al., 2009). Recent work expanded this concept to encompass sequences of dynamically changing training criteria (Wang et al., 2021). Inspired by human development, several studies have investigated whether structuring visual experience along developmental trajectories can improve learning outcomes (Elman, 1993; Bengio et al., 2009; Vogelsang et al., 2018; Jinsi et al., 2023; Jang & Tong, 2021; Avberšek et al., 2021). However, these studies relied on labeled supervision, leaving unresolved how such curricula affect unsupervised or contrastive learning paradigms.

Contrastive learning offers a promising framework for understanding learning during this part of the lifespan because they do not require explicit labels to drive learning. By learning to identify similarities and differences between unlabelled visual inputs, contrastive models acquire generalizable and robust representations (He et al., 2020; Chen et al., 2020) through mechanisms that loosely resemble self-organized infant learning.

More recently, Sheybani et al. (2023) demonstrated that contrastive models trained on age-ordered infant visual data outperform those trained with random or reversed sequences, suggesting that the temporal progression of natural visual experience may facilitate representation learning. These results suggest that the progression of visual experience through the childhood lifespan, may play a functional role in learning. Although these observations have inspired developmental analogies in machine learning, it remains unclear which aspects, if any, translate into practical benefits for artificial systems. In principle, performance gains achieved through simple re-ordering of training data or other low-cost curriculum interventions would offer an appealing and computationally efficient strategy for improving learning outcomes.

Building on this insight, our work undertakes a more systematic investigation of several parameters that may characterize a natural developmental trajectory encountered by young children, and reflect a subset of the physics of sampling visual information from the world across different ages. Rather than presenting different image sets within each training phase, we use the same base images in early and later phases of the learning curriculum, but modify them in ways that mimic some aspects of naturally occurring changes in visual experience with age. Our goal is not to treat biological development as a normative blueprint for artificial learning, nor to overlook the substantial mechanistic differences between biological and artificial systems. Rather, we view developmental inspiration as a source of hypotheses that can be empirically tested. Given the increasing use of developmental narratives in motivating curriculum design, it is important to examine whether commonly assumed perceptual factors indeed contribute to improved representation learning. We define a developmental curriculum as a sequence of training inputs that progresses along a specific factor in a manner inspired by human developmental trajectories (e.g., blur-to-clear imagery, slow-to-fast movement, simple-to-complex lighting, or scene complexity). The hypothesis is that such a curriculum can accelerate learning and enhance the generalization ability of contrastive learning frameworks within schedule strategies examined here, even when applied to datasets that do not follow a developmental progression.

Infant visual perception matures alongside physiological and environmental changes, including increased mobility (Thelen et al., 1996; Adolph & Joh, 2006), growing environmental complexity (Thelen & Smith, 1994; Adolph & Berger, 2006), and improvements in visual clarity (Kellman & Arterberry, 2000). We therefore investigate whether explicitly incorporating these parameters into a dynamic, structured curriculum during pretext training can guide models along a developmental learning trajectory, promoting more effective representation learning within a constrained computational budget.

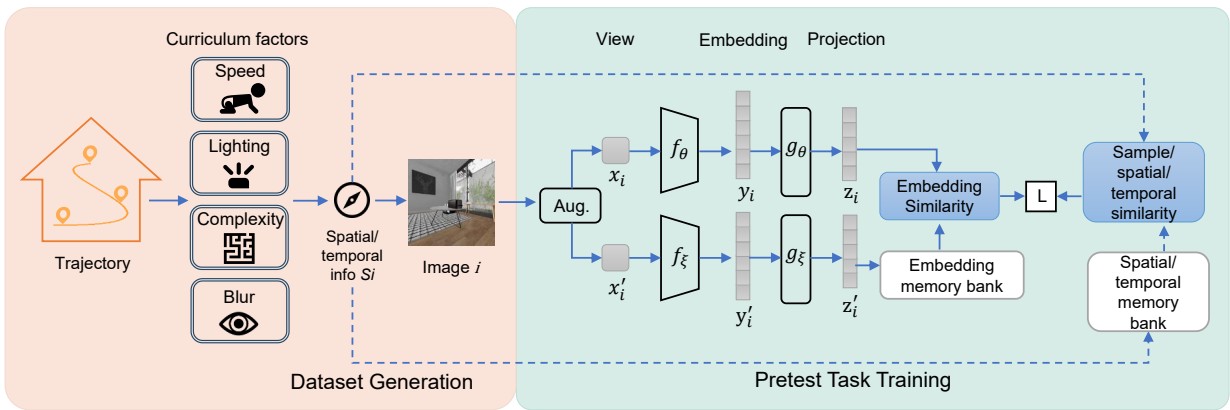

Figure 1: Overview of the contrastive learning framework with curriculum learning strategy. The four parameters in double-bordered boxes are dynamically adjusted according to the curriculum schedule. The contrastive approach calculates feature similarity between a key image and samples stored in the memory bank, encouraging higher similarity for related samples and lower similarity for unrelated ones.

Fig. 1 summarizes our framework. We pretrained visual contrastive learning models on realistic ray-traced images derived from an avatar traversing a virtual environment. Building on this framework, we systematically varied parameters of the image sampling process. We first controlled three dynamic parameters, image blur, lighting complexity, and avatar movement speed, by applying Gaussian blur, increasing lighting

variability within the virtual environment, and adjusting the avatar's temporal sampling rate, respectively. The performance of the pretrained foundation model was assessed using image classification accuracy on the ImageNet (Deng et al., 2009) and Toybox (Wang et al., 2017) datasets. We defined the learning sequence that aligns with putative human developmental trajectories as the *developmental mode*, and its reverse as the *anti-developmental mode*. Experimental results showed that although there were improvements based on increasing the variability of lighting overall, introducing a developmental learning curriculum did not enhance the quality of the learned representations. Generally, both developmental and anti-developmental sequences yielded highly similar results, and curriculum learning showed minimal benefit relative to baseline conditions.

We next examined the role of image complexity as proposed in Sheybani et al. (2023). Unlike their approach, which balanced image complexity across stages and eliminated curriculum-induced gains, we divided the dataset into simple and complex subsets and trained the model sequentially. Unexpectedly, models trained in a *complex-to-simple* order outperformed those trained in the *simple-to-complex* order on downstream tasks. This trend was further validated using the real-world SAYCam dataset (Sullivan et al., 2021), where the same pattern was observed.

These findings challenge the assumption that human developmental progression, at least insofar as it relates to the physics of visual experience, is inherently advantageous for representation learning. Our results instead highlight the need to critically reassess the role of developmental analogies in machine learning and to design curriculum mechanisms grounded in empirical evidence rather than presumed biological parallels.

## 2 Related Work

### 2.1 Developmentally Inspired Visual Curricula: Blur and Spatial Frequency

Curriculum learning has been shown to improve training efficiency when task difficulty is aligned with a meaningful progression (Bengio et al., 2009). Building on this general principle, a series of studies have examined curricula informed by developmental trajectories in human visual experience.

Several computational studies have specifically probed the role of blur or spatial frequency manipulation, key aspects of early infant vision, in shaping visual learning. Vogelsang et al. (2018) demonstrated that training convolutional networks with a coarse-to-fine schedule, beginning with blurred images and progressing to sharper ones, led to receptive fields that integrate information over larger spatial extents and improved generalization across image resolutions. This finding was interpreted as evidence for a functional advantage of coarse-to-fine visual experience, consistent with developmental evidence that infants initially resolve only lower spatial frequencies (Dobson & Teller, 1978). Building on this idea, Jinsi et al. (2023) systematically varied the initial blur level and its reduction over training, reporting that CNNs first exposed to low-pass filtered (blurred) inputs learned basic-level categories more rapidly and transferred more effectively to subordinate-level tasks than models trained exclusively on clear images. In contrast, Avberšek et al. (2021) examined curricula based on the removal of high spatial frequencies and found that although a coarse-to-fine regime improved performance on blurred test images, it did not yield superior classification on unblurred images, highlighting limitations of blur-based curricula in standard classification settings. Similarly, Jang & Tong (2021) observed that a blurry-to-clear schedule enhanced robustness to blur in face recognition tasks but not in generic object recognition, indicating task specificity in how blur influences learned representations. These studies used supervised training with explicit labels, an assumption that does not reflect early human development, leaving unresolved how unsupervised or self-supervised learning paradigms may be affected differently by curricula.

### 2.2 Naturalistic Developmental Curriculum

Using contrastive learning, Sheybani et al. (2023) defined curriculum progression by ordering a training set of images collected by forehead-mounted cameras according to the increasing age of children used to collect the samples. They showed that models trained in an age-sorted developmental sequence outperform those trained with random or reversed-order (i.e., anti-developmental) curricula. Moreover, a control analysis

further demonstrates that the benefit diminishes when the temporal sampling rate or the spatial complexity of visual information is fixed across the different sampling epochs. These results suggest that the progression of visual experience through the childhood lifespan, may play a functional role in learning. However, the content of those stimuli evolves substantially over the course of this developmental span, which obscures the cause of the benefit.

A recent preprint (Lu et al., 2025) reported that curriculum learning based on artificial manipulations of blur, contrast, and color complexity within an image set increased the model's shape bias and improved classification accuracy following image degradation. However, it remains unclear if these effects translate to improvements on general image classification tasks, and no comparison was made against a baseline condition.

## 3 Preliminaries

### 3.1 Momentum Contrastive (MoCo) Learning

We adopt MoCo v2 (He et al., 2020) as one of our contrastive learning baselines. As shown in Fig. 1, a key image $i$ is randomly augmented into two views $x_i$ and $x_i'$. They are separately encoded by a query encoder $f_\theta$ and a momentum encoder $f_\xi$ to get representations $y_i$ and $y_i'$, which are then projected into embedding $z_i$ and $z_i'$ through the corresponding MLP projection head $g_\theta$ and $g_\xi$. $z_i'$ is recorded in the memory bank. Feature similarity is calculated using cosine similarity. The contrastive objective encourages a view $x_i$ to align with its augmented positive counterpart $x_i'$ relative to all negatives:

$$L_i = -\log \frac{\exp(\mathrm{sim}(z_i, z_i')/\tau)}{\sum_{k \in M} \exp(\mathrm{sim}(z_i, z_k')/\tau)} , \tag{1}$$

where $M$ represents the memory bank. $\mathrm{sim}$(u,v) represents the cosine similarity between two vectors, and $\tau$ is the temperature parameter.

### 3.2 Temporal Contrastive Learning Model (Temp-MoCo)

Temp-MoCo extends MoCo by treating temporally adjacent frames as positive pairs. Following the implementation in (Orhan et al., 2020), for each key image, we randomly sample its immediate neighbors within a $\pm 1$-frame window (a 0.4-second temporal window at 5 fps). All other frames are treated as negatives. This encourages models to exploit natural temporal continuity in visual experience.

### 3.3 Environmental Spatial Similarity (ESS) Contrastive Learning

In human visual perception, certain neurons exhibit selectivity for objects while maintaining invariance to transformations such as changes in size, position, and rotation. Such invariances may emerge from the natural temporal contiguity of visual experiences (Li & DiCarlo, 2008) or from gradual variations in input features over time (Wood & Wood, 2018). Building upon this principle, the Environmental Spatial Similarity (ESS) approach (Zhu et al., 2022; 2024) extends contiguity-based learning to environment transformations to learn invariances that mirror those developed through human perceptual experience.

ESS computed image similarity based on the spatial locations where images are captured within the environment. The position distance between two samples, $\Delta P_{i,j}$, is calculated using Euclidean distance. The rotation angle was converted from quaternion representation to the yaw direction angle $r$, since the avatar rotated only within the yaw plane during data collection. The rotation distance between image $i$ and image $j$ is defined as:

$$\Delta R_{i,j} = \min(|r_i - r_j|, 360 - |r_i - r_j|) . \tag{2}$$

The loss function encouraged higher feature similarity between views sampled from spatially proximal locations in the environment. Two views were considered a positive pair if both their positional and rotational differences fell within predefined thresholds. This was formalized using an indicator function:

$$F_{\theta_P, \theta_R}(i, j) = \mathbb{I}(\Delta P_{i,j} < \theta_P \text{ and } \Delta R_{i,j} < \theta_R) , \tag{3}$$

which can be abbreviated as $F(i, j)$. The position and rotation thresholds, $\theta_P$ and $\theta_R$, were set to 0.8 meters and 12 degrees, respectively. The loss function for image $i$ was defined as the average loss across all positive pairs:

$$L_i = -\frac{1}{T} \sum_{j=0}^{j \in M} F(i, j) \log \frac{\exp(\text{sim}(z_i, z_p')/\tau)}{\sum_{k \in M} \exp(\text{sim}(z_i, z_k')/\tau)} \ , \tag{4}$$

where $T = \sum_{j \in M} F(i, j)$ denotes the total number of positive pair.

## 4 Methods

### 4.1 Pipeline Overview

Our approach integrates curriculum learning into self-supervised contrastive representation learning. First, images are processed under a curriculum schedule that progressively adjusts the visual properties of the input data (see section 4.3). Then, the resulting images are passed through a contrastive learning framework to learn representations. After pretext training, the encoder is frozen and a linear classifier is trained on top of the learned representations $f_\theta$ for downstream evaluation.

### 4.2 Basic Pretext Datasets

**Virtual datasets.** The House100K and House100KLighting datasets (Zhu et al., 2024) are generated using the ThreeDWorld (TDW) simulation platform (Gan et al., 2020) within the "Archviz House" environment, enhanced with 48 additional objects. A human-controlled avatar navigates the environment through translations, small jumps, and yaw rotations, producing a 102,196-step trajectory. At each step, it captures a $224 \times 224$ egocentric image, along with position $p_i$ and orientation $r_i$ (as quaternions).

In subsequent runs, the same trajectory is rendered under different lighting conditions. In the House100K dataset, all images are captured under the default lighting condition of TDW. The House100KLighting dataset selects nine representative skyboxes from 95 candidates, each introducing variation in the directionality and spectral characteristics of the light source, based on t-SNE (Hinton & Roweis, 2002) embeddings, with one skybox chosen from each region of a notional $3 \times 3$ grid. This setup allows control evaluation of models under systematic lighting variations while keeping all other visual factors constant.

**Real-world dataset.** The SAYCam dataset comprises real-world egocentric video data recorded using head-mounted cameras worn by three infants in their natural environments (Sullivan et al., 2021). For our work, only data from a single infant, referred to as S, is used, consisting of approximately two hours of weekly video recordings collected between 6 and 30 months of age. Following the procedure described in Orhan et al. (2020), videos are sampled at 5 frames per second, resulting in approximately 2.9 million images for the pretext task. Each frame is resized such that its shorter side measures 256 pixels, followed by a $224 \times 224$ center crop. The resulting dataset is denoted as SAYCam-S.

### 4.3 Curriculum Design in Contrastive Pretraining

To simulate human-like developmental learning in contrastive representation learning, we design curriculum learning strategies that dynamically adjust four visual factors during pretext training: image blur, movement speed, lighting conditions, and scene complexity. Each factor is modulated in either a developmental or anti-developmental mode, corresponding respectively to progressions that mimic or invert the assumed trajectory of typical infant perceptual development.

#### 4.3.1 Image Blur

To simulate the developmental trajectory of visual clarity in infants, we progressively adjust the radius parameter $R$ of Gaussian-kernel over pretext-training epochs. The schedule is:

$$R = \begin{cases} 223 - 2\lfloor 0.85 \log_{1.04}(200 - i) \rfloor, & \text{if } i < 170 \\ 0, & \text{if } i \geq 170 \end{cases}, \tag{5}$$

where $i$ denotes the effective epoch index: $i$ equals the current epoch in the developmental mode, and $i = 200 - \text{epoch}$ in anti-developmental mode. As shown in Fig. 2, image clarity increases with training in the developmental mode and decreases in the anti-developmental mode. To minimize confounding variables, we disable random Gaussian-blur augmentation in all blur-related experiments.

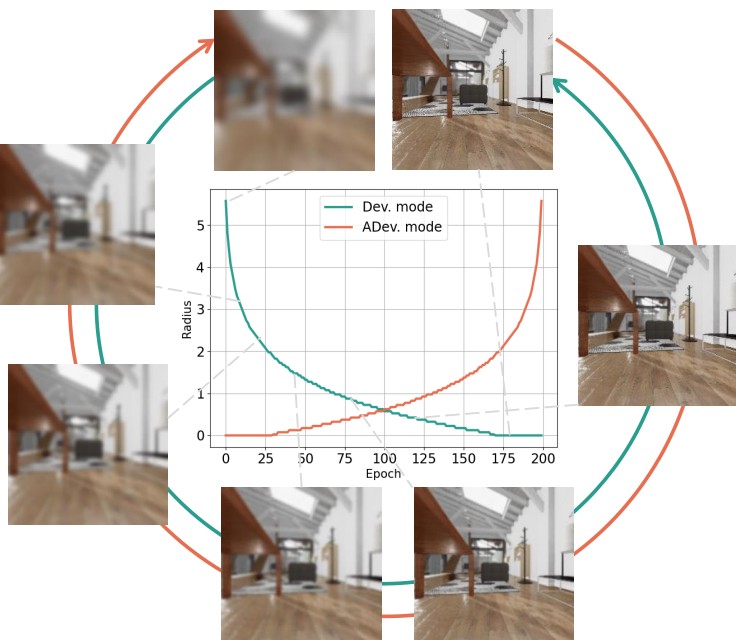

Figure 2: Gaussian blur schedule and visual examples across training. Two curves show the relationship between Gaussian blur radius and training epoch. The surrounding images illustrate how blur progressively decreases in the developmental mode, with images (arranged counterclockwise from the top) corresponding to epochs 0, 10, 20, 40, 80, 120, and 200.

### 4.3.2 Movement Speed

To simulate developmental changes in movement speed, we construct a series of datasets by manipulating the sampling frequency of the original datasets, although this manipulation does not directly correspond to the actual movement speed during recording. Conceptually, faster movement corresponds to a lower temporal sampling density of visual input, whereas slower movement is associated with a higher sampling density.

For the virtual dataset House100K, each image represents a visual fixation at a specific location within the house. We interpolate additional frames between fixations to simulate slower movement (House200K), and downsample frames to simulate faster movement (House50K and House25K), providing a total of four datasets of different sizes. Positional coordinates are linearly averaged, and rotations are smoothed using quaternion-based spherical linear interpolation (Slerp) (Shoemake, 1985). For more details, please refer to Appendix H.

SAYCam-S is initially sampled at 5 frames per second. To simulate faster movement, we create two additional datasets with frames downsampled by factors of two and four, corresponding to progressively sparser temporal sampling.

In the developmental mode, the dataset sizes used in the phases range from large to small to simulate faster and faster movement speeds, as shown in Fig 3. In the anti-developmental mode, the same datasets are used, but in reverse order. To ensure a consistent total training volume across stages and settings, the number of training epochs per stage is scaled up or down accordingly.

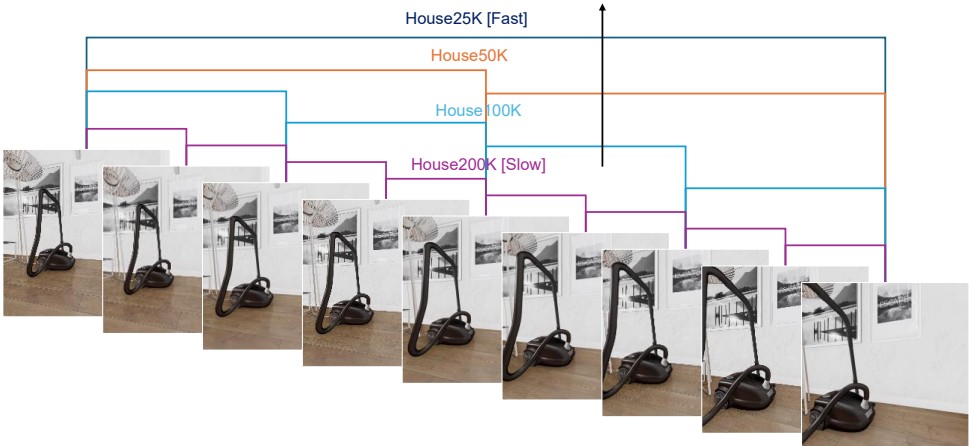

Figure 3: Comparison across datasets with different speeds. Developmental training proceeds from bottom to top, progressively moving from slower to faster.

### 4.3.3 Lighting Conditions

Lighting conditions are chosen to partially represent the complexity of the environment. In particular, varying the lighting provides a controlled way to introduce visual diversity across scenes without changing the content of images and surfaces, as demonstrated by the strong benefit of lighting on downstream performance in prior work Zhu et al. (2024). Due to the characteristics of the dataset, this setting was only conducted on the House100KLighting. The nine skyboxes are roughly ranked by their similarity to the default lighting condition according to visual inspection of representative images, as shown in Fig. 4. This ordering provides a gradual progression of lighting complexity, enabling controlled variation during training to investigate how lighting diversity influences model learning.

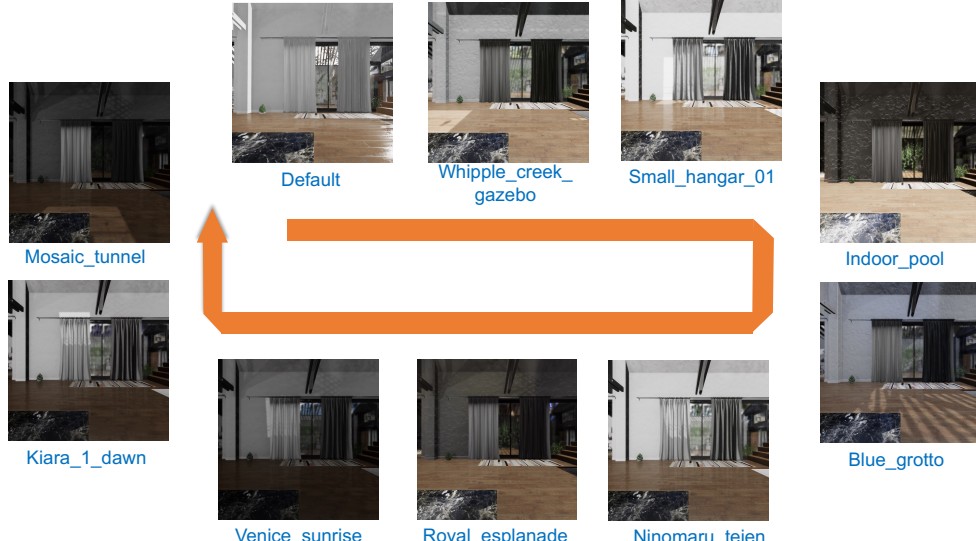

Figure 4: Image samples captured at the same position and rotation under different lighting conditions. The skyboxes are arranged as whipple_creek_gazebo, small_hangar_01, indoor_pool, blue_grotto, ninomaru_teien, royal_esplanade, venice_sunrise, kiara_1_dawn and mosaic_tunnel, ordered from most to least similar to the default lighting condition.

Pretraining is divided evenly into ten stages. In the developmental mode, training begins with the default skybox, and at each subsequent stage, an additional skybox is introduced following the predefined order, resulting in a progressively expanding set of lighting conditions. For each image, the data loader randomly assigns one lighting condition to each of the two augmented views, sampled from the current available set. If only a single skybox is present at a given stage, both views are rendered under the same condition. In the anti-developmental mode, training begins with all ten lighting models, and one is removed at each stage in reverse order.

### 4.3.4   Scene Complexity

To assess the effect of image complexity on representation learning, we follow the method proposed in Sheybani et al. (2023), which quantifies complexity as the proportion of edge pixels detected by the Canny edge detector (Canny, 1986) relative to the total number of pixels in an image. We apply the Canny detector with a lower threshold of 100, an upper threshold of 200, and a Sobel kernel size of 3. Under these settings, we process all images in the House100K and SAYCam-S datasets and compute the distribution of edge ratios, as shown in Fig. 5. Based on the median of the corresponding distribution, each dataset is evenly divided into two subsets: simple scenes (low edge ratio) and complex scenes (high edge ratio). For House100K, the mean edge ratios for the simple and complex subsets are $0.073 \pm 0.00010$ and $0.139 \pm 0.00015$, respectively. For SAYCam-S, the mean edge ratios for the simple and complex subsets are $0.027 \pm 0.00003$ and $0.104 \pm 0.00010$, respectively. We acknowledge that this split is a simplification. However, these results demonstrate that the simple and complex subsets exhibit systematic differences in edge density, capturing meaningful distinctions in image complexity rather than merely subtle variations.

For curriculum learning, we divide the pretext training epochs into two equal phases. In the developmental mode, the model is trained sequentially on the simple subset followed by the complex subset, emulating an assumed increase in perceptual complexity during early development. In the anti-developmental mode, this order is reversed.

## 5   Experiment and Results

### 5.1   Summary

In this section, we evaluate the effects of four developmental curriculum factors: image blur, movement speed, lighting conditions, and scene complexity. Each factor was manipulated in both developmental and anti-developmental orders, with baseline models trained on early- and late-stage configurations for comparison. Experiments were first conducted in a virtual environment (House100K and House100KLighting datasets) and then validated on a real-world dataset (SAYCam-S), particularly for movement speed and scene complexity, allowing us both to test generality and to extend prior findings from infant-inspired curriculum experience.

Across all tested conditions, we find no consistent benefit of the factor-wise developmental curricula relative to stable training. Marginal improvements in some settings (e.g., blur or lighting) are not statistically significant, and complexity-ordered curricula even underperform the anti-developmental mode in some cases. For completeness and to assess generality, we report additional results using Bootstrap Your Own Latent (BYOL) Grill et al. (2020) on the movement speed and scene complexity factors for the House100K dataset in the Appendix C, which show patterns consistent with those observed for the other models.

### 5.2   Pretrain Settings

We used MoCo and ESS as the base models for the virtual dataset, and Temp-MoCo for the real-world dataset. We dynamically adjusted parameters analogous to key variables in human developmental progression. When these parameters evolved according to the presumed trajectory observed in human learning, we defined the configuration as the *developmental mode*; when altered in the opposite direction, it was termed the *anti-developmental mode*. For comparison, the *early- and late-stage baseline* models were trained using the parameter settings corresponding to the beginning and end of the developmental sequences, respectively.

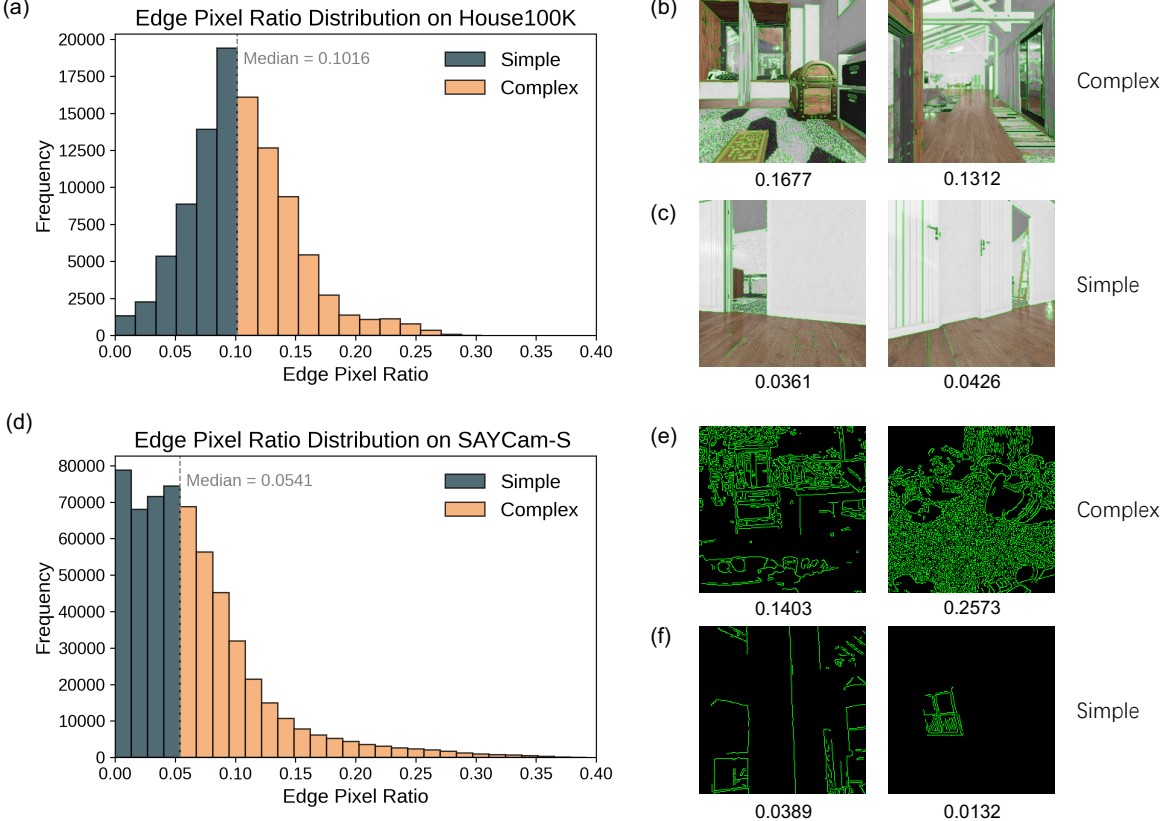

Figure 5: Edge-based image complexity in the House100K and SAYCam-S datasets. (a) Distribution of edge ratios in House100K, computed using Canny edge detection. The vertical dashed line indicates the median value used to divide the dataset into simple and complex subsets. (b,c) Representative samples from the high-complexity (complex) and low-complexity (simple) groups in House100K with detected edges highlighted in green. The number below each image indicates its edge ratio (complexity index). (d) Distribution of edge ratios in SAYCam-S. (e,f) Canny edge detection results for representative samples from the complex and simple groups in SAYCam-S.

Unless otherwise specified, each configuration was trained for 200 epochs on the virtual dataset and repeated three times. The results were averaged. Detailed training configurations are provided in Appendix A.

### 5.3 Downstream Task Settings

We evaluated model performance on the ImageNet (Deng et al., 2009) and Toybox (Wang et al., 2017) classification tasks. The Toybox dataset contains 12 toy object categories, each with 30 individual instances. For each instance, video sequences were provided under multiple transformations, including object present, absent, hodgepodge, and various translations and rotations. We followed the data sampling procedures of Orhan et al.. All videos were recorded at 30 frames per second. We used the first 27 instances from each category for training and the remaining 3 for evaluation. From each video, excluding absent sequences that contain no objects, we sampled one frame every 5 frames to construct the image dataset. The model was trained to classify each input image into one of the 12 object categories, with input images randomly shuffled across categories during training.

### 5.4 Blur Curriculum in Virtual Environment

As shown in Fig. 6, there was no curricular benefit of blurring for either ESS or MoCo training when compared against training on an unblurred dataset. While the developmental training mode for blur with the ESS model achieved higher classification accuracy in the developmental mode than in the anti-developmental mode, with improvements of 8.71% on ImageNet and 7.45% on Toybox, the developmental sequence did not improve accuracy relative to the training set *late*, which used exclusively unblurred images. The MoCo model results were similar. Thus, there was no evident advantage for training either model with a blur to clear gradient in image quality.

Table 1: Training arrangement for developmental and anti-developmental mode of movement speed.

| Dev. epoch | [0,25) | [25,75) | [75,175) | [175,375) |
|---|---|---|---|---|
| Dataset size | 200K | 100K | 50K | 25K |
| Ant. epoch | [0,200) | [200,300) | [300,350) | [350,375) |
| Dataset size | 25K | 50K | 100K | 200K |

### 5.5 Movement Speed Curriculum in Virtual Environment

For the simulation of changes in movement speed, again we observe no measurable benefit of this curriculum formulation, corresponding to a reduction in the data set sizes from House 200K to House 25K. First, we measured how accuracy changes with the size of the data set, as shown in Fig. 7 and observed that classification accuracy decreases slightly as dataset size decreases, but overall performance remains comparable across conditions.

We next applied curriculum learning strategies by sequentially presenting datasets of varying simulated movement speeds during pretext training as shown in Table 1. Fig. 6 shows that the developmental curriculum achieved slightly higher average accuracies than the anti-developmental counterpart for both ESS and MoCo models across two downstream tasks. However, these differences were not statistically significant, suggesting that both curriculum strategies yield comparable performance. Moreover, both curriculum-based results were on par with those obtained using the most detailed dataset, House200K, indicating that the introduction of a curriculum did not confer additional benefits under this setup.

### 5.6 Lighting Conditions Curriculum in Virtual Environment

As children age, they experience an increasingly diverse range of visual environments, which was simulated here by varying the lighting conditions used to generate images. We first trained models with a cosine learning rate schedule on datasets rendered under the default lighting condition from the simulation platform and under each of nine additional skyboxes, separately. As shown in Fig. 7, performance on the ImageNet

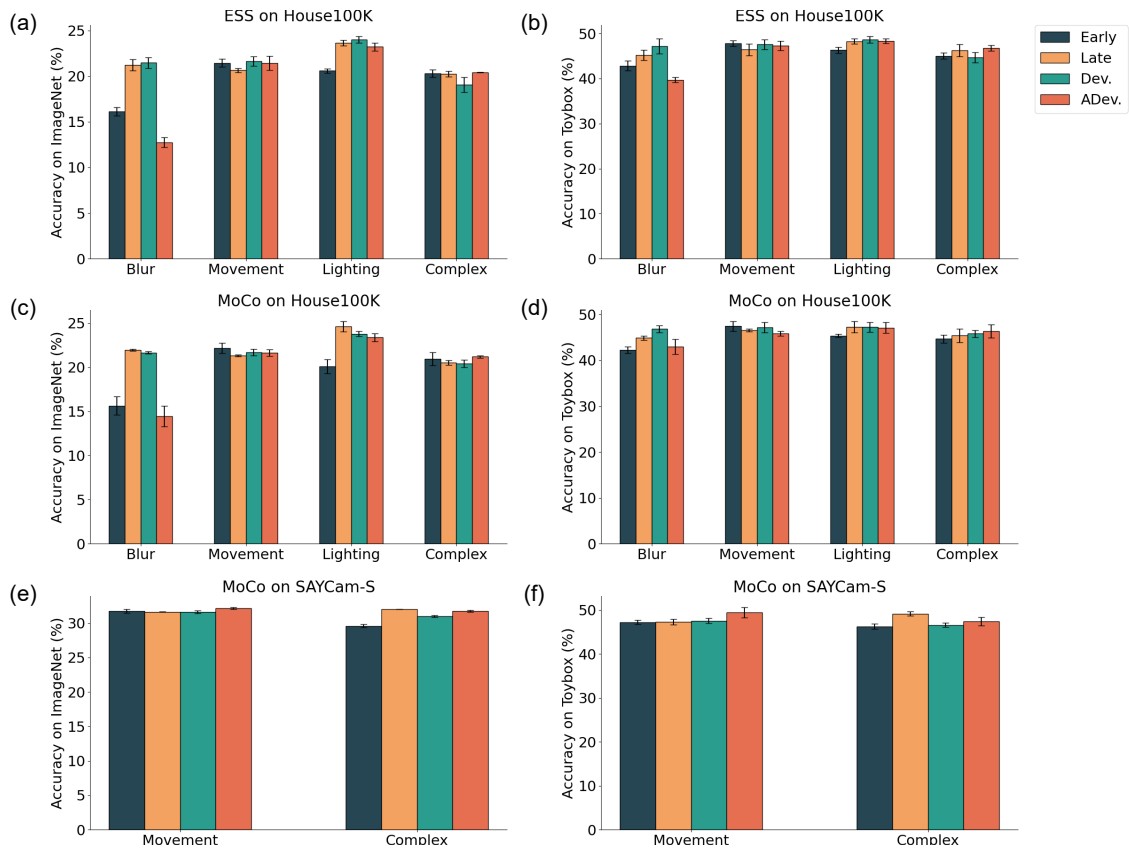

Figure 6: Comparison of downstream accuracy under different curriculum learning conditions. The left column shows results on ImageNet, and the right column shows results on Toybox. (a,b) ESS trained on House100K. (c,d) MoCo trained on House100K. (e,f) MoCo trained on SAYCam-S. Bars indicate mean accuracy; error bars show standard deviation across trials. Detailed numerical results are provided in the Appendix B.

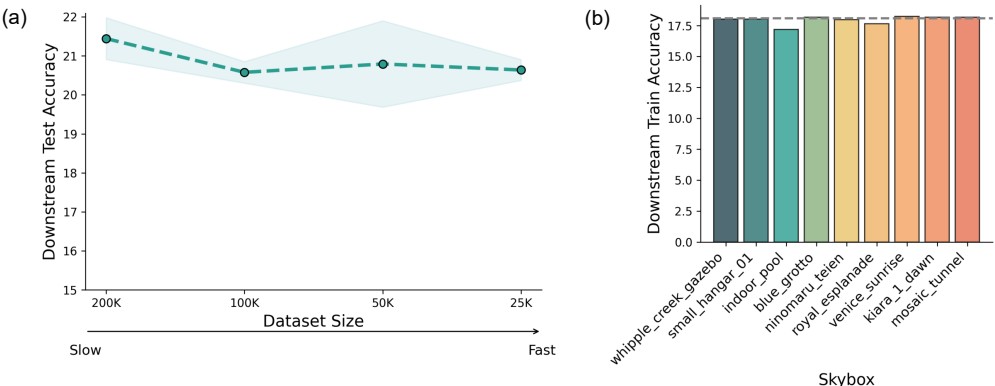

Figure 7: Downstream ImageNet accuracy of ESS under various movement speed and lighting conditions as curriculum learning baselines. (a) ESS trained on House200K, House100K, House50K, and House25K, with decreasing dataset size reflecting increasing avatar movement speed. (b) ESS trained on datasets incorporating nine additional skyboxes to vary lighting complexity. The gray dashed line indicates performance on House100K with the default lighting environment.

classification task for each of the nine skyboxes closely matched that obtained under the default lighting, ranging from 95.09% to 100.83% of default accuracy.

To simulate developmental and anti-developmental curricula, we sequentially added or removed one skybox every 20 epochs according to their manually defined similarity to the default lighting. The dataset with only the default lighting and the dataset containing all ten lighting conditions (default plus nine skyboxes) served as the early- and late-stage baselines, respectively. As shown in Fig. 6, the effect of lighting complexity paralleled those observed for movement speed, with no clear difference between the developmental or anti-developmental conditions for either ESS or MoCo training. Moreover, both curriculum strategies achieve comparable performance to the model trained for 200 epochs across all ten lighting conditions.

## 5.7 Image Complexity Curriculum in Virtual Environment

We examined whether visual scene complexity influences the effectiveness of curriculum learning, motivated by the findings by Sheybani et al. (2023) which showed that selecting a complexity-matched subset from the original dataset across developmental stages removes the advantage of age-ordered input sequences. To this end, the data were split into simple and complex subsets according to Canny edge image complexity. In the developmental mode, the model was trained first on the simple subset and then on the complex subset, whereas in the anti-developmental mode, the order was reversed.

As shown in Fig. 6, the ESS model trained with the developmental curriculum yielded the lowest downstream classification accuracy on both ImageNet and Toybox, performing 1.36% and 2.05% worse, respectively, than the anti-developmental mode. Models trained exclusively on either the simple or complex subsets, which are non-curriculum baselines, resulted in performance comparable to the anti-developmental curriculum, with differences that were not statistically significant. A similar trend was observed for the MoCo model. These results suggest that, under our operational definition of visual complexity, curriculum learning along the complexity dimension may even impair representation quality rather than enhance it.

## 5.8 Movement and Complexity Curricula in Realistic Environment

In Sheybani et al. (2023), the superiority of visual inputs from younger age groups was attributed to the slowness and simplicity of their early visual experience. This aligns with our baseline findings: under the same training budget, training on the slow movement condition (House200K) outperformed fast condition (House25K), and training on simple images yielded better results than training on complex ones. However, unlike (Sheybani et al., 2023), our curriculum learning experiments did not exhibit such superiority.

To further investigate this discrepancy, we replicated the experiments on a real-world dataset, SAYCam-S, using Temp-MoCo as the pretext learning framework. The model was trained for 12 epochs on SAYCam-S as the baseline budget and we evaluated models on the same downstream tasks, with minor adjustments to the experimental settings. This additional evaluation was designed to test whether curriculum learning under dynamic conditions would yield any benefits when trained on more diverse and naturalistic visual inputs.

**Movement.** SAYCam-S was used as the baseline pretext dataset representing slow movement, with training conducted for 12 epochs. A fast movement baseline was created by selecting every fourth frame from SAYCam-S and training for 48 epochs, thereby maintaining an equivalent total training budget. Under the same pretext training budget, the developmental mode used frame sampling at every frame, every second frame, and every fourth frame for 4, 8, and 16 epochs, respectively. The anti-developmental mode used the same configuration in reverse order. The complexity-based curriculum followed the same method as described earlier. As shown in Fig. 6, model performance across the fast, slow, and developmental modes was highly similar. The anti-developmental schedule produced slightly higher accuracies than the other three modes, although these differences were not statistically significant.

**Complexity.** We quantified image complexity of SAYCam-S using the Canny edge detection method. To examine whether complexity varied with developmental stages, we visualized the distributions of image complexity across three age stages (6–14 months, 15–21 months, and 22–30 months) in Fig. 8. The distributions revealed no clear progression from simple to complex scenes over time. We then divided SAYCam-S into

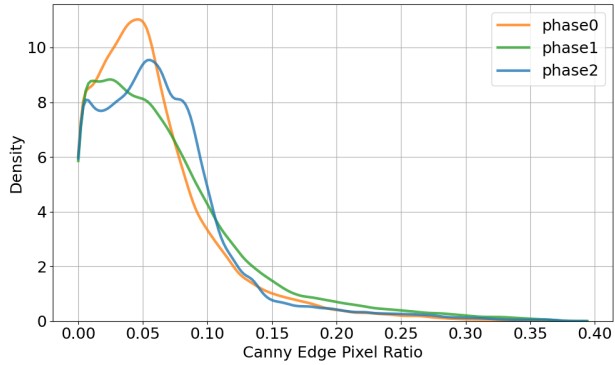

Figure 8: Complexity distributions of SAYCam-S across three phases.

simple and complex subsets based on the median complexity threshold, as shown in Fig. 5, and conducted curriculum learning experiments using these subsets.

Overall, SAYCam-S exhibited lower complexity than House100K, and the image density remained nearly constant in the low-complexity region. In the downstream tasks, as shown in Fig. 6, models trained on the complex subset achieved markedly better performance, followed by those trained in the anti-developmental mode, whereas the developmental mode and on the simple set of images performed worse. These results suggest that the models preferentially benefited from the more complex visual inputs in SAYCam-S, contrary to the assumption that developmental progression from simple to complex enhances learning.

## 6 Discussion

### 6.1 Limited Benefits of Developmental Curricula under Tested Settings

We investigated whether introducing developmental visual input that mimic aspects of human perceptual maturation can improve contrastive learning, as measured by downstream classification performance. Across the four parameters we evaluated, blur, movement speed, and lighting, although the developmental curriculum yielded marginally higher accuracies than the anti-developmental mode in some experiments, these differences were not statistically reliable and remained comparable to baseline models, indicating no additional benefit from curriculum learning. Notably, for the complexity-based curriculum, the developmental mode consistently underperformed relative to the anti-developmental mode, indicating that increasing scene complexity, defined here by edge density, may have hindered, rather than facilitated, representation learning.

These findings imply that while certain features of human developmental trajectories can inspire curriculum design, their effects are difficult to capture through a limited set of controllable variables on universal datasets.

### 6.2 Discrepancy with Prior Work

Prior studies that focused on single-factor curricula, such as blur-to-clear schedules (Vogelsang et al., 2018; Jinsi et al., 2023; Avberšek et al., 2021; Jang & Tong, 2021), generally rely on supervised training with explicit labels, which are unavailable to infants. Moreover, the reported benefits of blur-based curricula were limited and highly task-specific: while performance on blurred inputs improved, this often came at the expense of accuracy on clear images. Our work examines contrastive learning and evaluates multiple factors, showing that the tested single-factor curricula do not yield consistent improvements under the given training regimes and evaluation suite.

Sheybani et al. (2023) used dynamically evolving naturalistic infant data and attempted to identify which aspects of the changing visual experience drive learning benefits. We fixed the training set and manipulated individual factors to isolate their causal influence. Our results suggest that factors highlighted by Sheybani

et al. (2023), including movement speed and environmental complexity, may not represent fundamental drivers of developmental curriculum benefits.

Lu et al. (2025) combined multiple visual factors—such as acuity, contrast, and color—within a single curriculum, whereas we isolated individual factors to control their influence. Furthermore, their focus was on achieving near-human levels of shape bias, rather than evaluating general classification performance. We also assessed shape bias in models pretrained on House100K but did not observe meaningful improvements under single-factor curricula (Appendix F). Taken together, these discrepancies suggest that developmental-inspired curricula may not confer universal gains in representation learning, and that empirical evaluation of individual factors is crucial for understanding their functional impact.

### 6.3  Possible Factors Affecting Curriculum Effectiveness

Several factors may account for the limited or even detrimental effects observed in our experiments.

#### 6.3.1  Optimal Sequencing of Training Samples

First, the optimal sequencing of training samples remains a subject of debate (Wang et al., 2021). Curriculum learning is classically defined as presenting samples from easy to difficult, with difficulty typically determined by human-designed metrics such as complexity (Wei et al., 2016) or diversity (Bengio et al., 2009). In contrast, hard example mining (Shrivastava et al., 2016) adopts the opposite approach, deliberately prioritizing the most challenging samples early in training. This raises a fundamental question: is an easy-to-hard or hard-to-easy progression more advantageous? Different strategies may foster different learning dynamics.

#### 6.3.2  Definition of Sample Difficulty

Defining "difficulty" in curriculum learning, when modeled after human visual development, remains ambiguous for pretext training. A natural intuition is to use pretext loss as a proxy for difficulty, with higher loss corresponding to harder samples and lower loss to easier ones. Under this definition, blurred images would be easier than sharp ones, and samples involving slower movements would be easier than those with faster dynamics (see Appendix B). However, this ordering does not align with the trajectory of human perceptual development. Only variations in lighting conditions, progressing from few to many, can be plausibly interpreted as following an easy-to-difficult continuum under this definition.

#### 6.3.3  Impact of Hard Negatives in Contrastive Learning

In contrastive learning, hard negatives (i.e., samples that are close in the feature space but do not constitute positive pairs) are known to enhance discriminability and improve representation quality (Robinson et al., 2020), even though they increase the loss value. This further demonstrates that pretext loss is not a reliable indicator of sample difficulty. To further investigate, we evaluated the Bootstrap Your Own Latent model (BYOL) (Grill et al., 2020), a contrastive learning model that does not rely on negative samples, and observed very limited improvement caused by the developmental curriculum (Appendix C), indicating that the influence of hard negatives might not be the primary factor.

#### 6.3.4  Mismatch Between Pretext and Downstream Tasks

Another possibility is that the benefits of curriculum learning may not manifest primarily in standard evaluation metrics such as object classification accuracy. More broadly, the landscape of pretext tasks may not correspond directly to the objectives of downstream tasks, creating a potential mismatch between curriculum design and downstream performance. Prior work has suggested that developmental curricula can promote shape-based rather than texture-based representations (Lu et al., 2025). To compare with these results, we evaluated the shape bias of our pretrained models (see Appendix F), but found no consistent enhancement or systematic shift attributable to curriculum structure.

### 6.3.5 Simulation Design and Curriculum Scheduling Limitations

Finally, the simulated environment and curriculum schedules we employed may not have sufficiently reflected realistic developmental progression. The selection of factors, such as motion speed and lighting variation, along with their scheduling, may not provide an optimal simulation for evaluating curriculum learning.

### 6.3.6 Experimental Settings and Computational Constraints

The observed limited effects may also reflect constraints in experimental settings. Dataset diversity, memory bank sizes, and pretraining budgets could influence downstream performance. For example, House100K contains images from a single environment, and SAYCam-S covers only the first months of visual experience, inherently limiting absolute accuracy. In Appendix E, we report additional experiments with an enlarged memory bank and extended pretraining, which yield consistent findings. Nevertheless, the null results may still be conditioned on computational constraints, and larger-scale evaluations remain an important direction for future work.

### 6.4 Limitations and Future Work

Beyond these explanatory factors, the present work has several methodological limitations that point to clear opportunities for future improvement. These gaps between our approximation of a subset of developmental variables and the experience of actual children means that we cannot conclude from these experiments that there are no benefits of developmentally inspired curriculums. First, more cognitively and physically grounded definitions of curriculum variables should be explored. For example, slower movement speed in our simulations was modeled through positional interpolation, which increased the number of images sampled within a given region but did not capture the true motion of objects in a physical environment. Similarly, visual complexity was estimated using Canny edge detection, which provides only a coarse approximation of scene complexity (see Appendix G). Future work could incorporate more realistic object motion and richer scene attributes such as object count, spatial density, or texture density.

Second, curriculum schedules could be designed adaptively rather than manually. Adaptive curriculum methods, such as self-paced learning (Jiang et al., 2015), which selects samples based on model performance, and reinforcement learning teacher (Matiisen et al., 2019), which dynamically adjust data scheduling, offer a more responsive mechanism for aligning training progression with the model's evolving learning capacity.

Third, broader and more naturalistic datasets should be employed. Pretraining on datasets that capture a wider range of real-world scenes, lighting conditions, and egocentric dynamics, such as Ego4D (Grauman et al., 2022), may provide a more ecologically valid setting in which curriculum strategies can reveal their potential benefits.

Finally, future studies could examine alternative self-supervised learning objectives beyond contrastive frameworks. Methods such as MAE (He et al., 2022), which do not explicitly rely on negative sampling, may interact differently with curriculum-based input structuring and merit systematic exploration.

## 7 Conclusion

We systematically evaluated developmentally inspired curricula for contrastive visual representation learning across multiple controlled visual factors, including blur, lighting, movement speed, and scene complexity. The results show no consistent benefit of the specific factor-wise developmental schedules considered here relative to stable training, and complexity-ordered curricula even underperform anti-developmental mode. These findings suggest limitations on intuitive mappings from infant visual development to effective training curricula in self-supervised learning within the evaluated contrastive learning frameworks.

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

## A    Implementation Details

### A.1    Settings of Virtual Dataset Training

All models were trained on three NVIDIA RTX A6000 GPUs using a ResNet-50 backbone encoder. The pretext task was trained for 200 epochs with a batch size of 192. Due to the relatively small dataset size, the memory bank was limited to 4096 entries. Each configuration was trained three times, and results were averaged, except for the single lighting condition experiments, where each skybox setting was trained once without repetition.

Unless otherwise specified, a fixed learning rate of 0.3 was used across all models to maintain a consistent update rate regardless of variations in input samples at different curriculum stages. A cosine learning rate schedule (initialized at 0.3), was used only in the single lighting condition experiments.

Data augmentation included random resized cropping to $224 \times 224$ with a scale range of (0.2, 1.0), color jittering applied with an 80% probability, random grayscale conversion with a 20% probability, and Gaussian blur and horizontal flipping both applied with a 50% probability. For experiments using blur level as the curriculum variable, random Gaussian blur was omitted from the set of image augmentations to avoid confounding effects. All other hyperparameters, such as the temperature parameter, and the momentum and the weight decay values for the SGD optimizer, followed the original MoCo V2 implementation unless otherwise noted.

For downstream evaluation, only the final linear layer was trained for 50 epochs. Training was conducted on either three NVIDIA RTX A6000 GPUs with batch size 192 or four NVIDIA V100 GPUs with batch size 256. SGD was used with momentum 0.9. The learning rate was initialized at 30, reduced to 3 at epoch 30, and further decreased to 0.3 at epoch 40. Evaluation on the test set was performed every 5 epochs.

### A.2 Settings of Real-world Training

For pretext training, We used a ResNext-50 (Xie et al., 2017) encoder as the backbone network, initialized with random weights. The output dimension of the projection head was set to 128, and the memory bank size was 65,536. The model was trained for 12 epochs using SGD with a fixed learning rate of 0.015. All experiments were conducted with a batch size of 192 on three NVIDIA RTX A6000 GPUs.

In the downstream tasks, the settings for the two classification experiments followed those described earlier. We fine-tuned a linear classifier on frozen features extracted by the trained encoder, following the protocol in Orhan et al. (2020). The classifier was trained for 25 epochs using the Adam optimizer with a learning rate of 0.0005, which was decayed by a factor of 0.2 at epochs 23 and 24. We conducted training on three NVIDIA RTX A6000 GPUs or four NVIDIA V100 GPUs.

## B Detailed Quantitative Results

This section presents detailed quantitative results corresponding to the experiments on House100K (Table 2, and Table 3) and SAYCam-S (Table 4) in the main text.

## C BYOL Results on Movement Speed and Scene Complexity

To further assess the generality of our findings across learning frameworks, we conducted additional experiments using BYOL on the movement speed and scene complexity curriculum factors for the House100K dataset. These factors were selected because prior developmental datasets and analyses have particularly emphasized their role in shaping visual experience.

We followed the same experimental protocol as in the main text, evaluating developmental and anti-developmental curricula alongside stable training baselines. The results, summarized in Table 5, show patterns consistent with those observed for the other models: we do not find reliable advantages of developmental ordering on movement speed and complexity, and performance differences across curriculum conditions remain small and inconsistent.

### C.1 Implementation Details

BYOL uses a ResNet-50 backbone. The projection head consists of a two-layer MLP with a hidden dimension of 4096 and an output projection size of 256. The online network is trained with stochastic gradient descent (SGD) using a learning rate of 0.03, momentum of 0.9, and weight decay of $4 \times 10^{-4}$. Training is performed with a batch size of 256 for 200 epochs. The target network parameters are updated using an exponential moving average with momentum coefficient $m = 0.996$.The settings of the two downstream tasks are the same as those used in ESS and MoCo.

Table 2: Curriculum learning results across three factors on House100K. ES and LS denote the early-stage and late-stage baselines, respectively. Dev. represents the developmental mode. ADev. represents the anti-developmental mode. "Acc." indicates the classification accuracy on the corresponding downstream task at the end of 50 epochs. Values following "$\pm$" represent the standard error of the mean.

| Condition | Model | Mode | Pretrain Training loss ↓ | Downstream Test acc.(%) ImageNet | Toybox |
|---|---|---|---|---|---|
| Blur | ESS | ES (blur) | $4.25 \pm 0.002$ | $16.12 \pm 0.32$ | $42.82 \pm 0.77$ |
| | | LS (clear) | $4.21 \pm 0.006$ | $21.21 \pm 0.45$ | $45.20 \pm 0.81$ |
| | | Dev. | $4.21 \pm 0.004$ | $\mathbf{21.46} \pm 0.43$ | $\mathbf{47.16} \pm 1.81$ |
| | | ADev. | $4.30 \pm 0.003$ | $12.75 \pm 0.38$ | $39.70 \pm 0.39$ |
| | MoCo | ES (blur) | $4.02 \pm 0.002$ | $15.63 \pm 0.75$ | $42.27 \pm 0.49$ |
| | | LS (clear) | $3.94 \pm 0.002$ | $\mathbf{21.96} \pm 0.08$ | $44.91 \pm 0.33$ |
| | | Dev. | $3.95 \pm 0.005$ | $21.64 \pm 0.10$ | $\mathbf{46.83} \pm 0.56$ |
| | | ADev. | $4.03 \pm 0.002$ | $14.45 \pm 0.84$ | $42.99 \pm 1.17$ |
| Movement | ESS | ES (slow) | $4.24 \pm 0.006$ | $21.44 \pm 0.31$ | $\mathbf{47.84} \pm 0.45$ |
| | | LS (fast) | $4.22 \pm 0.003$ | $20.64 \pm 0.15$ | $46.41 \pm 0.90$ |
| | | Dev. | $4.22 \pm 0.011$ | $\mathbf{21.63} \pm 0.36$ | $47.57 \pm 0.77$ |
| | | ADev. | $4.24 \pm 0.004$ | $21.43 \pm 0.56$ | $47.29 \pm 0.71$ |
| | MoCo | ES (slow) | $3.98 \pm 0.006$ | $\mathbf{22.16} \pm 0.41$ | $\mathbf{47.45} \pm 0.75$ |
| | | LS (fast) | $3.95 \pm 0.002$ | $21.32 \pm 0.07$ | $46.54 \pm 0.24$ |
| | | Dev. | $3.96 \pm 0.002$ | $21.69 \pm 0.27$ | $47.20 \pm 0.80$ |
| | | ADev. | $3.98 \pm 0.003$ | $21.62 \pm 0.27$ | $45.84 \pm 0.34$ |
| Lighting | ESS | ES (default) | $4.22 \pm 0.007$ | $20.58 \pm 0.16$ | $46.32 \pm 0.49$ |
| | | LS (10 lightings) | $4.32 \pm 0.005$ | $23.63 \pm 0.22$ | $48.28 \pm 0.40$ |
| | | Dev. | $4.30 \pm 0.001$ | $\mathbf{23.99} \pm 0.06$ | $\mathbf{48.65} \pm 0.52$ |
| | | ADev. | $4.24 \pm 0.005$ | $23.22 \pm 0.28$ | $48.33 \pm 0.40$ |
| | MoCo | ES (default) | $3.97 \pm 0.003$ | $20.19 \pm 0.48$ | $45.35 \pm 0.26$ |
| | | LS (10 lightings) | $4.09 \pm 0.003$ | $\mathbf{24.42} \pm 0.30$ | $46.91 \pm 0.50$ |
| | | Dev. | $4.08 \pm 0.006$ | $23.93 \pm 0.32$ | $\mathbf{47.25} \pm 0.75$ |
| | | ADev. | $4.00 \pm 0.002$ | $23.37 \pm 0.31$ | $47.12 \pm 0.85$ |

Table 3: Curriculum learning results across image complexity on House100K.

| Condition | Model | Mode | Pretrain Training loss ↓ | Downstream Test acc.(%) ImageNet | Toybox |
|---|---|---|---|---|---|
| Complexity | ESS | ES (simple) | $4.35 \pm 0.005$ | $20.28 \pm 0.28$ | $45.05 \pm 0.47$ |
| | | LS (complex) | $4.28 \pm 0.006$ | $20.25 \pm 0.21$ | $46.26 \pm 0.95$ |
| | | Dev. | $4.27 \pm 0.003$ | $19.06 \pm 0.58$ | $44.68 \pm 0.81$ |
| | | ADev. | $4.36 \pm 0.002$ | $\mathbf{20.42} \pm 0.01$ | $\mathbf{46.73} \pm 0.45$ |
| | MoCo | ES (simple) | $4.08 \pm 0.002$ | $20.93 \pm 0.52$ | $44.67 \pm 0.63$ |
| | | LS (complex) | $4.05 \pm 0.004$ | $20.51 \pm 0.19$ | $45.38 \pm 1.05$ |
| | | Dev. | $4.05 \pm 0.003$ | $20.40 \pm 0.30$ | $45.80 \pm 0.54$ |
| | | ADev. | $4.07 \pm 0.002$ | $\mathbf{21.20} \pm 0.09$ | $\mathbf{46.40} \pm 1.02$ |

## D Evidence of Converged Training

To address concerns about potential underfitting, we analyzed the loss reduction in the final pretraining epochs. For House100K, the loss change is computed by comparing the reduction over the last 10 pretext epochs with that over the full 200 pretraining epochs, or a similar training volume. For example, if the dataset

Table 4: Curriculum learning results across movement speed and complexity of MoCo on SAYCam-S.

| Condition | Mode | Pretrain Training loss ↓ | Downstream Test acc.(%) ImageNet | Toybox |
|---|---|---|---|---|
| Movement | ES (slow) | $7.12 \pm 0.001$ | $31.64 \pm 0.03$ | $47.31 \pm 0.43$ |
| | LS (fast) | $7.27 \pm 0.002$ | $31.74 \pm 0.17$ | $47.26 \pm 0.36$ |
| | Dev. | $7.30 \pm 0.003$ | $31.58 \pm 0.14$ | $47.57 \pm 0.41$ |
| | ADev. | $7.09 \pm 0.003$ | $\mathbf{32.15} \pm 0.10$ | $\mathbf{49.44} \pm 0.85$ |
| Complexity | ES (simple) | $7.28 \pm 0.009$ | $29.59 \pm 0.17$ | $46.29 \pm 0.40$ |
| | LS (complex) | $6.89 \pm 0.003$ | $\mathbf{32.06} \pm 0.06$ | $\mathbf{49.18} \pm 0.33$ |
| | Dev. | $7.00 \pm 0.001$ | $30.98 \pm 0.11$ | $46.59 \pm 0.33$ |
| | ADev. | $7.24 \pm 0.003$ | $31.72 \pm 0.10$ | $47.37 \pm 0.68$ |

Table 5: Curriculum learning results across movement speed and complexity of BYOL on House100K.

| Condition | Mode | Pretrain Training loss ↓ | Downstream Test acc.(%) ImageNet | Toybox |
|---|---|---|---|---|
| Movement | ES (slow) | $0.200 \pm 0.001$ | $\mathbf{24.98} \pm 0.05$ | $49.49 \pm 0.97$ |
| | LS (fast) | $0.182 \pm 0.001$ | $23.85 \pm 0.34$ | $48.26 \pm 1.02$ |
| | Dev. | $0.182 \pm 0.001$ | $24.33 \pm 0.12$ | $\mathbf{50.30} \pm 1.14$ |
| Complexity | LS (complex) | $0.143 \pm 0.001$ | $23.11 \pm 0.07$ | $47.85 \pm 0.70$ |
| | Dev. | $0.140 \pm 0.001$ | $\mathbf{23.70} \pm 0.06$ | $\mathbf{48.68} \pm 0.59$ |

is House200K, then the number is the ratio of the reduction of the last 5 epochs over the total 100 epochs,. For SAYCam-S, the loss change is computed by comparing the reduction over the last 2 epochs with that over the full 24-epoch training schedule. As shown in Table 6, for House100K, the loss change in these final epochs is minimal compared to the overall training reduction. The results indicate that the models trained on House100K have effectively converged, confirming that underfitting is not an issue. For SAYCam-S, the final two pretext epochs show a small decrease relative to the full 24-epoch training. Therefore, in Appendix E.2, we additionally report results with doubled pretraining volume to examine whether insufficient convergence could impact the outcomes.

Table 6: Loss reduction comparison in final epochs (ES: early stage, LS: last stage, Dev.: developmental, ADev.: anti-developmental). All values are in percentage (%).

| Condition | Dataset | Model | ES | LS | Dev. | ADev. |
|---|---|---|---|---|---|---|
| Blur | House100K | ESS | $0.10 \pm 0.03$ | $0.06 \pm 0.09$ | $0.05 \pm 0.03$ | $-1.60 \pm 0.01$ |
| | | MoCo | $-0.01 \pm 0.08$ | $-0.03 \pm 0.11$ | $-0.00 \pm 0.10$ | $-1.44 \pm 0.01$ |
| Movement | House100K | ESS | $0.04 \pm 0.06$ | $0.04 \pm 0.08$ | $0.14 \pm 0.20$ | $0.01 \pm 0.03$ |
| | | MoCo | $0.09 \pm 0.02$ | $0.24 \pm 0.21$ | $0.22 \pm 0.13$ | $0.18 \pm 0.02$ |
| | | BYOL | $-0.11 \pm 0.09$ | $0.12 \pm 0.09$ | $-0.03 \pm 0.04$ | N/A |
| | SAYCam-S | MoCo | $1.41 \pm 0.02$ | $1.65 \pm 0.04$ | $1.68 \pm 0.08$ | $1.34 \pm 0.00$ |
| Lighting | House100K | ESS | $0.05 \pm 0.12$ | $0.04 \pm 0.05$ | $0.71 \pm 0.54$ | $0.34 \pm 0.05$ |
| | | MoCo | $0.12 \pm 0.12$ | $0.05 \pm 0.04$ | $0.09 \pm 0.14$ | $1.16 \pm 0.09$ |
| Movement | House100K | ESS | $0.17 \pm 0.13$ | $0.13 \pm 0.09$ | $0.11 \pm 0.12$ | $0.15 \pm 0.08$ |
| | | MoCo | $0.34 \pm 0.11$ | $0.36 \pm 0.10$ | $0.26 \pm 0.03$ | $0.33 \pm 0.08$ |
| | | BYOL | N/A | $0.01 \pm 0.03$ | $0.08 \pm 0.04$ | N/A |
| | SAYCam-S | MoCo | $1.58 \pm 0.04$ | $1.33 \pm 0.03$ | $2.29 \pm 0.01$ | $1.70 \pm 0.01$ |

# E    Additional Results for MoCo under Larger-scale Extensions

We report additional experimental results for MoCo under modified pretraining configurations on House100K and SAYCam-S.

## E.1    MoCo on House100K: Memory Bank Size of 65,536

To evaluate the effect of memory bank size, we increased the queue size to 65,536 while keeping all other settings identical to the original pretext training configuration, including the number of training epochs.

Due to computational constraints, we evaluated this configuration only under the movement speed and complexity factors. As shown in Table 7, the results are highly consistent with those reported in the main text, showing comparable performance trends and confirming that our conclusions are robust to this change in memory bank size.

Table 7: Curriculum learning results of MoCo on House100K with the memory bank size of 65,536.

| Condition | Model | Mode | Pretrain Training loss ↓ | Downstream Test acc.(%) ImageNet | Toybox |
|---|---|---|---|---|---|
| Movement | MoCo | ES (simple) | $6.69 \pm 0.003$ | $\mathbf{22.09} \pm 0.16$ | $47.39 \pm 0.78$ |
| | | LS (complex) | $6.70 \pm 0.004$ | $21.31 \pm 0.37$ | $46.06 \pm 0.52$ |
| | | Dev. | $6.69 \pm 0.002$ | $21.66 \pm 0.21$ | $47.31 \pm 0.86$ |
| | | ADev. | $6.71 \pm 0.005$ | $21.55 \pm 0.35$ | $\mathbf{47.43} \pm 0.74$ |
| Complexity | MoCo | ES (simple) | $6.82 \pm 0.050$ | $20.78 \pm 0.24$ | $45.78 \pm 0.73$ |
| | | LS (complex) | $6.81 \pm 0.006$ | $20.58 \pm 0.21$ | $45.76 \pm 0.74$ |
| | | Dev. | $6.82 \pm 0.003$ | $20.36 \pm 0.25$ | $44.45 \pm 1.01$ |
| | | ADev. | $6.82 \pm 0.018$ | $\mathbf{20.88} \pm 0.29$ | $\mathbf{47.30} \pm 0.95$ |

## E.2    MoCo on SAYCam-S: Longer Pretraining Setting

To further assess the impact of extended pretraining, we doubled the pretrain epoch of complexity on SAYCam-S to 48. Experiments under the complexity factor for the late-stage, developmental, and anti-developmental modes are conducted. For each setting, we only performed two runs.

As shown in Table 8, the anti-developmental setting performs slightly better than the developmental mode, but does not outperform fixed training on the complex dataset. This suggests that extended pretraining does not alter our overall conclusion that developmental curriculum learning of complexity on SAYCasm-S does not improve downstream performance.

Table 8: Curriculum learning results across the complexity of MoCo on SAYCam-S with doubled training volume.

| Condition | Mode | Pretrain Training loss ↓ | Downstream Test acc.(%) ImageNet | Toybox |
|---|---|---|---|---|
| Complexity | LS (complex) | $6.70 \pm 0.002$ | $\mathbf{34.38} \pm 0.28$ | $49.57 \pm 0.48$ |
| | Dev. | $6.77 \pm 0.001$ | $34.10 \pm 0.06$ | $49.15 \pm 0.44$ |
| | ADev. | $7.04 \pm 0.001$ | $34.28 \pm 0.11$ | $\mathbf{50.56} \pm 0.58$ |

# F    Shape Bias Results

The cue-conflict dataset consists of images that combine the shape of one object with the texture of another across 16 categories. It was designed to study human and neural network preferences for shape versus texture cues in visual recognition (Geirhos et al., 2018). We mapped the outputs of models trained on the

downstream ImageNet classification task from 1,000 classes to the 16 cue-conflict categories and then froze the models. The total number of images correctly classified to the shape $N_{shape}$ and texture $N_{texture}$ were counted. Following the methodology in Geirhos et al. (2018); Lu et al. (2025), we computed shape bias as:

$$B_{shape} = \frac{N_{shape}}{N_{shape} + N_{texture}} \ .$$ (6)

This metric quantifies the tendency of a visual recognition system to rely more on object shape or texture during classification. As illustrated in Fig. 9, the shape bias of models trained under the developmental and anti-developmental modes generally fell between those of the early- and late-stage baselines, indicating that curriculum learning along dimensions such as blur level, movement speed, lighting conditions, or scene complexity did not improve shape bias.

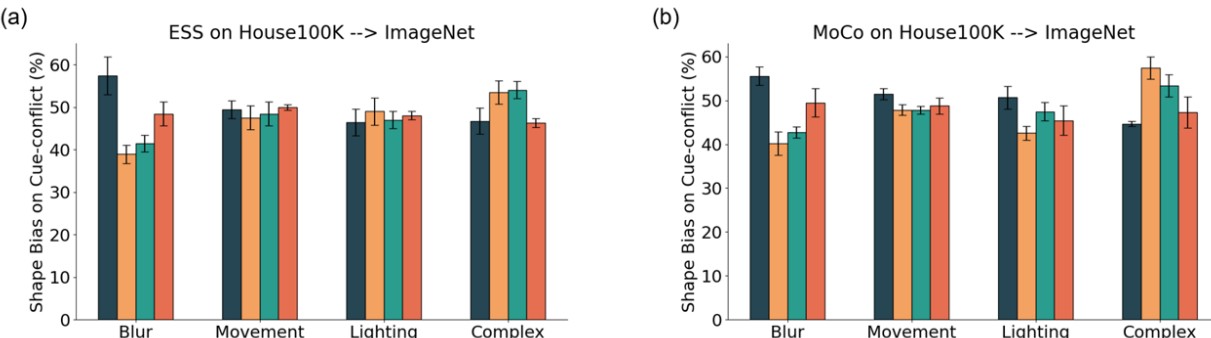

Figure 9: **Comparison of shape bias under different curriculum learning conditions.** (a) ESS backbone model. (b) MoCo backbone model.

## G    Definition of Complexity

The definition of image complexity remains contested. Rigau et al. framed it from an information-theoretic perspective, either as the number of partitions needed to capture a target information ratio or as compositional complexity via Jensen-Shannon divergence. Mahon & Lukasiewicz instead applied hierarchical clustering of image patches with the minimum description length (MDL) principle to separate "meaningful complexity" from noise.

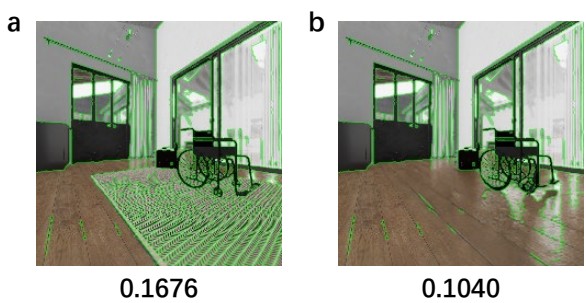

Figure 10: Limitations of edge-based complexity measures. Images with high edge density may primarily capture texture rather than perceptual complexity, highlighting that visual complexity cannot be reduced to a single low-level feature.

As shown in Fig. 10, high edge density may reflect surface texture rather than genuine perceptual complexity. This highlights the inherent difficulty of defining complexity, which cannot be attributed to any single visual

property but instead emerges from the interaction of multiple factors, including object density, structural organization, lighting variation, and semantic content. Consequently, visual complexity remains an elusive construct that resists precise quantification.

## H  Trajectory Smoothness and Interpolation Rationale

For interpolation, positional coordinates are linearly averaged, and quaternion-based spherical linear interpolation (Slerp) is used to smooth rotations and avoid gimbal lock. The typical distance between adjacent fixations ranged from 0 to 0.2 meters, and yaw rotation differences were generally below 5 degrees, making linear interpolation appropriate.

The trajectory of House100K dataset includes 102, 196 steps. To ensure the validity of interpolation for both position and rotation, we verify that the step-to-step differences are sufficiently small throughout the dataset. The dataset is recorded as a continuous egocentric sequence while an avatar travels through the environment. An exception occurs only at step 21,951, where the avatar was stuck between two flower pots and then returned to the initial point at step 21,952. The positional displacement between these two steps is 7.42 meters, while all other positional distances fall within a heavy-tailed distribution between 0 and 0.3 meters (Fig. 11 a), indicating that the avatar typically moves in small, smooth increments even though its trajectory is not strictly linear.

The distribution of rotational differences is more varied. The rotational force is applied manually. The resulting yaw changes occasionally accumulate to as much as 49.40 degrees. However, the vast majority of rotations are concentrated between 0 and 4 degrees, with a small number of larger peaks (Fig. 11 b). The peaks are likely related to the participants' movement habits. The large rotation differences are not dominant in the trajectory (Fig. 11 c). Since converting quaternions to Euler angles for interpolation can introduce gimbal lock, we apply Slerp to interpolate rotations directly in quaternion space.

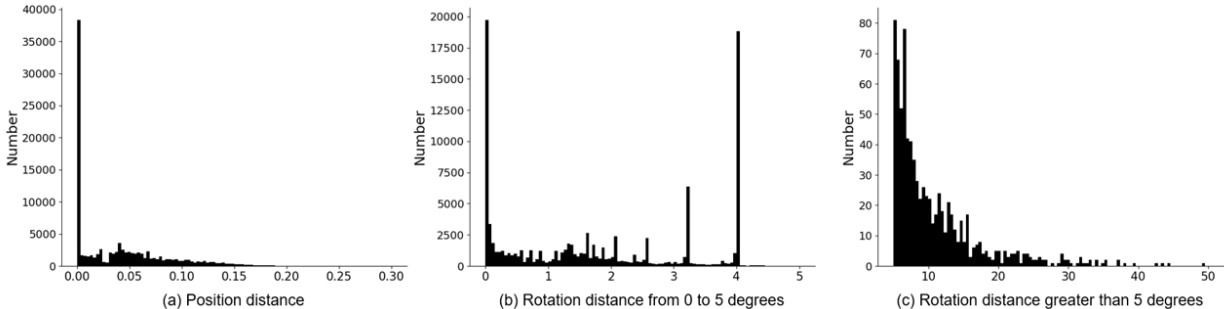

Figure 11: The distribution of position distance and rotation distance of House100K dataset.

