# OpenReview forum: "Rethinking Developmental Curricula for Contrastive Visual Learning"
_TMLR — Rejected by TMLR_

### Review · Reviewer_FJHG · 2025-12-16

**Summary Of Contributions:**

This paper presents a systematic and well-controlled investigation into whether developmental curricula improve contrastive visual representation learning. The authors evaluate four dynamic factors (image blur, movement speed, lighting complexity, and scene complexity) in both developmental (simple-to-complex) and anti-developmental (complex-to-simple) sequences across virtual and real-world egocentric datasets. The key finding is that none of the tested developmental curricula consistently improved downstream classification performance compared to stable training baselines.
Advantages: Methodologically rigorous, with careful control of variables and use of both synthetic and real data. Clear results that are important for the field, challenging a popular hypothesis.Well-written, with a thorough discussion of limitations and potential explanations for the findings
Limitations: The study is limited to contrastive learning frameworks; findings may not generalize to other self-supervised methods.

**Additional Comments:**

No additional comments

**Audience:**

Yes

**Audience Explanation:**

Its findings challenge a growing body of work that posits benefits from developmentally-inspired training sequences. Researchers in machine learning, cognitive science, and computer vision would find these results valuable for guiding future work.

**Claims And Evidence:**

Yes

**Claims Explanation:**

Extensive experiments across multiple datasets.

**Requested Changes:**

Clarify the operational definition of “developmental curriculum.” The paper interchangeably refers to “developmental” as both a specific sequence (e.g., blur-to-clear) and as a general analogy. A clear definition early in the paper (e.g., in Section 3 or 4) would help readers contextualize the claims.

Discuss the generalizability of findings beyond contrastive learning. The study is limited to MoCo-family models. A brief discussion on whether similar results might be expected for other self-supervised paradigms would strengthen the paper’s impact.

Minor:

Expand the discussion on why developmental curricula failed. The authors suggest several factors (e.g., difficulty metrics, negative samples), but a more structured synthesis, perhaps as a subsection in Section 6, would help readers better interpret the null results.

Include a visual example of the curriculum schedules in the main text, not just in the appendix, to improve readability and intuition.

Page 11, Figure 7 caption: “SAMCam-S” should be “SAYCam-S”; Ensure consistency. e.g.,  “Adev.” and “ADev.” in Appendix Table 2

---

> ### Author Response · Authors · 2026-02-26
> **We clarified the developmental curriculum definition, added generalizability tests, improved discussion structure, included visual examples, and corrected minor errors.**
>
> We sincerely thank the reviewer for the thorough and thoughtful evaluation of our manuscript. We appreciate the recognition of the methodological rigor of our study and the value of the null findings for the community. Below, we address each of the points raised.
>
> 1. **On the claim of a developmental curriculum:**
>    We note that the `developmental curriculum'' is used as a broader analogy to human perceptual development in the Introduction. We have modified it. In the fourth paragraph of the Introduction, we have added the definition of the developmental curriculum'' (marked as blue) before we propose our hypothesis.
>
> 2. **On generalizability:**
>    To further examine generalizability, we have added preliminary experiments with Bootstrap Your Own Latent (BYOL) in Appendix C. These additional results indicate that developmental curricula do not consistently improve performance in that setting either. Additionally, in Section 6.4. of the revised manuscript (highlighted in blue), we explicitly state that our conclusions may not extend to all contrastive learning variants, and we temper our claims accordingly.
>
> 3. **On the Synthesis and Structure of Curriculum Failure Discussion:**
>    We have modified Section 6.3. as a structured subsection with clear subheadings to clarify the potential reasons behind the null results.
>
> 4. **On the requested visual example:**
>    We have added Fig. 3 to visualize different datasets for various movement speeds and how they changed during the curriculum training. We also have modified Fig. 4 to present the developmental sequence. Now, each factor has a corresponding visualization showing how the training inputs evolve throughout the curriculum in the main text.
>
> 5. **Minor corrections:**
>    We have corrected `SAMCam-S'' to SAYCam-S'' in Figure 7 and standardized the abbreviation “ADev.” throughout the appendix. We also addressed several other minor inconsistencies.
>
> We thank the reviewer again for the constructive suggestions, which have meaningfully improved the clarity and presentation of our work.

---

### Review · Reviewer_mica · 2025-12-24

**Summary Of Contributions:**

The paper investigates whether developmental curriculum learning benefits contrastive visual representation learning. The authors modulate four factors—image blur, lighting conditions, movement speed, and scene complexity—within a virtual environment (House100K) and on a real-world infant egocentric dataset (SAYCam). By comparing learning curricula that follow order of developmental progressions against stable baselines and anti-developmental orders, the authors report that developmental curricula do not yield consistent downstream classification improvements compared to baselines, and in the case of complexity, the anti-developmental order sometimes performs better.

**Audience:**

Yes

**Audience Explanation:**

The effectiveness of developmental learning mechanisms remains an open problem in self-supervised learning. Reports of negative results are valuable to the community as they challenge intuitive assumptions and can prevent wasted research effort. If the limitations are addressed, clarifying boundaries on when and why developmental curricula fail would benefit researchers designing data-efficient learning algorithms.

**Claims And Evidence:**

No

**Claims Explanation:**

While the experiments provide negative results on some specific configurations tested, the broad conclusion that developmental curricula are ineffective is not fully supported due to potential methodological limitations. Specifically, the absolute performance of the models (e.g., linear probe accuracy) is relatively low, and the training budgets (e.g., small memory bank size of 4096; 12-48 epochs on SAYCam) may be insufficient to saturate the baseline models, potentially masking subtle curriculum effects that might emerge in a higher-capacity regime. Therefore, the evidence supports that these specific proxies under constrained training budgets do not help, but generalizing this conclusion remains difficult without stronger baselines and more robust definitions of the curriculum factors.

**Requested Changes:**

1. Please scope the claims to the tested settings and sharpen what is ruled out. The strongest supported claim is that these particular factor-wise curricula and schedules do not yield gains under the given training regimes and evaluation suite. The discussion should more explicitly separate this from stronger statements about “developmental curricula” in general.
2. It is recommended to provide evidence that the negative results persist under a less constrained training budget. The current memory bank size and training duration raise concerns that the models may fall in the underfitting regime, potentially masking curriculum effects. A scaling analysis or a run with standard MoCo settings would clarify whether the ineffectiveness is due to the curriculum or to insufficient model capacity.
3. Please address minor clarity and formatting issues to improve readability. For example, on page 10, "To examined" should be corrected to "To examine", and the capitalization of dataset names (e.g., "House200k" vs "House200K") should be consistent throughout the text.

---

> ### Author Response · Authors · 2026-02-26
> **We clarified the scope of our claims, detailed experimental settings and limitations, and addressed minor issues.**
>
> We thank the reviewer for the thoughtful and constructive feedback.
>
> 1. **On scope and claims:**
>    We appreciate the suggestion to more precisely delimit the scope of our conclusions. First, we added additional experiments using a negative-free contrastive framework, Bootstrap Your Own Latent (BYOL), in Appendix C.
>    Due to the time limitation, developmental schedules are only applied to part of the movement speed and image complexity. They similarly failed to yield any significant performance gains relative to baseline or anti-developmental schedules. These results suggest that the ineffectiveness of factor-wise curricula is not confined to MoCo-style contrastive learning, and support a broader, yet method-qualified claim that the developmental analogues described in this paper do not enhance several representative contrastive learning paradigms.
>
>    Second, we revised the manuscript to avoid language that could be interpreted as making broad negative claims about ``developmental curricula'' in general. The Introduction, the Conclusion, and the framing in the Discussion section have been refined to clarify that our findings pertain specifically to the factors and schedule strategies examined in this study. All such revisions are marked in blue in the revised manuscript for clarity.
>
> 2. **On experimental settings:**
>    We appreciate the reviewer's concerns regarding the low downstream accuracy and the possibility that constrained training budgets may obscure subtle curriculum effects. The House100K dataset contains images from a single house environment, and SAYCam-S covers only the first few months of an infant’s visual experience, so absolute downstream performance is necessarily limited by dataset diversity. To address potential concerns about underfitting, we include in Appendix D additional analyses comparing the loss reduction in the final training epochs with the loss reduction over the full pretraining schedule: for House100K, we compare the last 10 pretext epochs with the full 200-epoch training. These analyses indicate that the models reached approximate convergence under the given pretraining budgets, suggesting that the observed limited benefits are unlikely due solely to insufficient training.
>
>    However, we also understand the reviewer's concern about the limited memory bank size and the training epochs. We clarify the experimental settings below:
>
>    - **ESS on House100K:**
>       All settings follow the original ESS paper. The memory bank size of 4096 was jointly tuned with the positive-pair threshold to achieve optimal downstream performance. Due to time and computational constraints, we did not repeat these experiments with a larger memory bank size.
>
>    - **MoCo on House100K:**
>       We increased the memory bank size to 65,536 while keeping the pretext training epochs the same as in the original paper. Due to computational limitations, we only tested it on movement speed and the complexity, and gained very similar results (updated in Appendix E.1).
>
>    - **MoCo on SAYCam-S:**
>       The original work used SAYCam-S for contrastive pretraining, which we adopted as our baseline. We therefore followed the 12-epoch pretext training setting. We agree that evaluating performance under less constrained computational settings would provide additional insight into scaling behavior. To this end, we conducted additional experiments for the complexity factor under the late-stage, developmental, and anti-developmental modes, with two additional runs per setting using doubled pretraining epochs. The results show similar trends and lead to the same conclusion as the original experiments. The updated results are reported in Appendix E.2.
>
>    - **BYOL on House100K:**
>       In baseline experiments, we observed that extending pretext training beyond 200 epochs led to decreased downstream accuracy, likely due to overfitting.
>
>    Finally, we now explicitly noted in the revised manuscript (marked in blue) that our conclusions may be conditioned on the available computational resources. Larger-scale extensions (e.g., standard ImageNet-scale MoCo settings) remain an important direction for future work to further evaluate potential curriculum effects.
>
> 3. **Minor issues:**
>    We have corrected minor clarity and formatting issues, including grammatical errors and inconsistencies in the capitalization of dataset names. We appreciate the reviewer's careful attention to these details.

---

### Review · Reviewer_t2qQ · 2026-02-11

**Summary Of Contributions:**

The paper explores multiple curriculum learning strategies inspired by human visual development, i.e., visual development in infants and mimic the changes in visual experience with age, ranging from blurry to clear images, increasing lighting complexity, movement speed, and image complexity. These changes are simulated for training and their effect is evaluated on downstream classification performance. Models learned with curriculum learning are compared to stable training and different variations of the complexity order. The paper concludes that there is no consistent benefit of developmental schedules over stable training.

**Audience:**

No

**Audience Explanation:**

This is difficult to answer. Although there can also be a place for potential negative results, i.e., here not being able to show the benefit of developmental schedules, it is difficult to get some clear takeaways from the paper. It would be helpful if the presented experiments had a clearer structure and focus, and if the motivation for the chosen design choices for simulating developmental schedules were more clearly stated.

**Claims And Evidence:**

No

**Claims Explanation:**

Although the curriculum learning is inspired by infant visual development, there are many assumptions made which are unclear whether they hold. Further, how changes in visual experience are modeled leaves much room for variation, and this paper represents only one of them. Together, this makes it difficult to draw clear conclusions. Precisely:
- Is the conclusion of no benefit due to the choice of how the visual changes are modeled, or are there significant differences in how infants and neural networks learn, as claimed in the paper?
- I assume, just because children start with blurred vision, that does not mean that this helps/is necessary with/for learning, neither for humans nor neural networks.
- Paper builds upon the motivation that human perception is superior in adaptability; however, I do not see why infant visual development is taken as the main motivation to achieve this adaptability in neural networks.
- Simulating the increased diversity of visual environments by varying the lighting conditions seems very simplified.
- Modeling increased speed results in a decreased dataset size, which can have additional effects on the accuracy.
- The simple and complex scenes only vary slightly around the median. Unclear how to draw conclusions from these slight differences in a large portion of the image around the median. Also, having just two subsets for scene complexity seems like a significant simplification of the world.
- There is a discrepancy with previous work (section 5.7). Without explanation of the differences it is unclear from where this comes.

Overall, the paper presents many experiments across varying settings and configurations (an almost overwhelming number of variations), making it difficult to draw clear conclusions.

**Requested Changes:**

- Provide a clearer motivation for why infant development can be an inspiration for learning of artificial neural networks.
- Clearer motivation of the chosen simulation strategies, e.g., why lighting conditions relate to visual diversity. Are there other simulation possibilities?
Currently, it seems a very simplistic realisation of the visual development from which it is difficult to draw clear conclusions made in the paper.

---

> ### Author Response · Authors · 2026-02-26
> **Clarification of Developmental Motivation, Simulation Design, and Interpretative Scope**
>
> We thank the reviewer for the thoughtful and detailed evaluation. Below, we respond to the concerns raised and outline the corresponding revisions made to the manuscript.
>
> 1. **On the motivation and the role of infant development as inspiration:**
>
> Infant development has often been cited in prior literature as a conceptual motivation for curriculum learning, as it exemplifies naturally structured learning trajectories that evolve over time. Our contribution builds on this line of inquiry by isolating specific developmental proxies and evaluate them under controlled experimental conditions. We agree that human adaptability does not, in itself, imply that developmental processes in infants transfer directly to neural network training dynamics.
>
> In response, we have revised the Abstract and Introduction to soften any language that could be interpreted as suggesting a direct transfer from human developmental processes to artificial systems. The revised text clarifies that our objective is to empirically test whether such developmental proxies are effective in neural network training, rather than to assume their validity. We also added a brief discussion in the Introduction explaining why infant development has served as inspiration, while explicitly acknowledging the conceptual and mechanistic gap between biological and artificial learning systems.  All such revisions are marked in blue in the revised manuscript for clarity.
>
> 2. **On simulation choices:**
>
> We consider that any simulation of developmental factors necessarily involves abstraction and simplification. Our design aims to balance realism with experimental control, and it largely follows prior literature to facilitate meaningful comparison.
>
>    1. **Blur:** Blur schedules have been extensively studied in both developmental vision and machine learning. We adopt this established paradigm as a canonical proxy within the scope of this study. To provide additional context, we added a paragraph in Section 2 summarizing relevant prior work and clarifying how our implementation relates to existing approaches.
>
>    2. **Movement Speed:** As stated in Section 5.4 and Table 1, we adjust the number of training epochs across stages such that the total training volume remains constant. We have added sentences in the Method section to emphasize this design choice.
>
>    3. **Lighting:** We acknowledge that lighting variation captures only one aspect of environmental diversity. We chose this factor because it is directly controllable in the House100K environment and has been shown in the ESS paper to have strong impact on downstream performance. In Section 4.3.3., We added a clarifying sentence to explain the connection between lighting variation and visual diversity.
>
>    4. **Simple/complex split:** We acknowledge that the simple-vs.-complex division is an operational simplification. The median split provides a controlled manner to isolate complexity effects while maintaining comparable distributions. To address the reviewer’s concern, we added an analysis summarizing distributional differences between subsets, demonstrating that the resulting partitions differ systematically rather than only marginally.
>
> We also expanded the limitations part to explicitly state that these factors should be understood as operational proxies rather than comprehensive models of human developmental processes. All those revisions are marked in blue in the revised manuscript for clarity.
>
> 3. **On interpretation of conclusions:**
>
> We agree that multiple interpretations of the results are possible. In response, we have revised the manuscript to adopt a more intentionally cautious tone in explaining our conclusions. We now avoid overgeneralization and explicitly frame our interpretation as one plausible account among several alternatives. In the limitation part, we elaborate on potential shortcomings in the curriculum design that may have influenced the observed outcomes.
>
> 4. **On improving experimental structure and clarity:**
>
> We appreciate that many experiments may obscure key takeaways. To improve clarity, we added a concise summary of the main findings at the beginning of the Experimental section (Section 5) to outline the logic structure of the experiments and guide the reader through the progression of analysis (marked in blue).
>
> 5. **On discrepancy with prior work:**
>
> We added a new subsection (Section 6.2) to more explicitly discuss differences between our findings and prior studies. In particular, earlier work typically (1) varies multiple developmental factors simultaneously, (2) focuses on a single proxy (e.g., blur), or (3) analyzes parameter changes as they arise naturally in real-world datasets. In contrast, our study manipulates individual factors within a controlled and fixed dataset, enabling isolation of their independent effects while minimizing distributional confounds. This distinction in experimental design may account for differences in empirical conclusions.

---

### Decision · Action_Editor_f1tE · 2026-04-06

**Recommendation:** Reject

**Audience:**

Yes

**Audience Explanation:**

The paper addresses a question which is clearly within TMLR’s scope and valuable. The main issue raised by reviewers is in the clarity and communication of the experimental logic. As one reviewer states, "clarifying boundaries on when and why developmental curricula fail would benefit researchers designing data-efficient learning algorithms".

**Claims And Evidence:**

No

**Claims Explanation:**

The paper investigates whether "developmental curricula" can improve visual representation learning. Experiments are conducted on a naturalistic curriculum present in the SAYCam dataset, as well as on structured developmental curricula generated by ordering data (and augmentations) by complexity. Training on these curricula is compared against stable training, in which all conditions are presented at once, and the main finding is that none of the proposed curriculum strategies outperforms stable training.

The reviewers appreciated the value of this negative result. However, they found that the initial submission overstated some conclusions and was limited by the scale of the models used in the experiments. Several of these concerns were partially addressed during rebuttal (e.g., by running additional larger scale models), which reviewers noted positively.

Yet, several important concerns remain; in particular the experimental setup and the "complexity ordering" is still not sufficiently well justified, which limits the interpretation of the results. Also, general clarity and presentation of the experimental setup and results was noted as a weakness.

Although reviewers agreed that the empirical basis for the negative findings was strengthened in the revision, the post-rebuttal discussion converged on the view that the paper's motivation and overall significance still fall short of the bar required for TMLR.